# PRRSV Detection by qPCR on Serum Samples Collected in Due-to-Wean Piglets in Five Positive Stable Breeding Herds Following a Sow Mass Vaccination with a Modified Live Vaccine: A Descriptive Study

**DOI:** 10.3390/vetsci10040294

**Published:** 2023-04-15

**Authors:** Arnaud Lebret, Valérie Normand, Charlotte Teixeira Costa, Ingrid Messager, Pauline Berton, Mathieu Brissonnier, Théo Nicolazo, Gwenaël Boulbria

**Affiliations:** 1Porc.Spective Swine Vet Practice, ZA de Gohélève, 56920 Noyal-Pontivy, France; 2Rezoolution Pig Consulting Services, ZA de Gohélève, 56920 Noyal-Pontivy, France; 3Boehringer Ingelheim Animal Health France, Swine Bussiness Unit, 16, rue Louis Pasteur, 44119 Treillères, France

**Keywords:** swine, PRRSV, PCR, modified-live vaccination, monitoring

## Abstract

**Simple Summary:**

The purpose of this study was to describe vaccinal strain detection using qPCR on blood samples collected from due-to-wean piglets after a mass vaccination of their dams with a modified live vaccine in five positive stable herds with different management practices including external and internal biosecurity measures.

**Abstract:**

Data concerning PRRSV-1 vaccine virus strains dissemination within vaccinated sow herds are scarce. However, it is a big concern for swine practitioners when designing the PRRSV diagnostics strategy in vaccinated farms. At the same time, the possibility of vaccine virus transmission from sows to their offspring is important to have in mind in order to limit the risk of recombination between different PPRSV-1 modified live virus vaccine (MLV1) when both sows and piglets have to be vaccinated. This study was conducted in five PRRSV-stable breeding herds. The selected farms presented different characteristics regarding production parameters and biosecurity management practices in order to be, as much as possible, representative of French swine production herds. In four different batches following a sow mass vaccination with a PRRSV-1 modified live virus vaccine (ReproCyc^®^ PRRS EU, Boehringer Ingelheim, Ingelheim, Germany), we failed to detect the vaccine virus in due-to-wean piglets in all of the herds. This should mean that the dissemination of the vaccinal strain is a rare event, even just after a sow vaccination, at least for the vaccine tested in our study.

## 1. Introduction

Porcine Reproductive and Respiratory Syndrome Virus (PRRSV) is the aetiologic agent of PRRS, the most economically important disease of the swine industry [1,2]. However, it remains an ongoing challenge for practitioners to improve diagnostics and PRRSV control in swine herds. Indeed, knowing the herd status and infection dynamics are essential to design proper control measures such as the management of gilt introduction or more general health interventions. To achieve it, it is essential to implement an effective surveillance strategy aiming to evidence PRRSV circulation and shedding. For many years in North America, PRRSV stabilization protocols have been implemented [3]. Such protocols combine mass vaccination and an improvement in biosecurity measures. In France, since the 2000s, using vaccination at a whole-herd level (sows, post-weaning piglets, and sometimes fattening pigs) with a modified-live virus (MLV) vaccine, together with closure of the farm and a unidirectional pig and human flow, have been evaluated [4].

The success of such stabilization programs is based on the demonstration of the absence of viral circulation in the breeding herd.

Collecting blood from due-to-wean piglets has been widely used as a diagnostic sample to monitor PRRSV infection in a sow herd [5]. Recently, the American Association of Swine Veterinarians (AASV) reviewed the breeding herd classification system, including new diagnostic protocols for status definition [6]. Even if new sample types are considered, such as processing fluids or family oral fluids, the use of blood samples from due-to-wean piglets remains the pivotal sample type for classifying the herds. Moreover, this new classification takes into account the possibility of the detection of a vaccinal virus strain in piglets in the two weeks interval after a sow vaccination. Then, a grace period of two weeks is recommended after sow vaccination before testing the piglets so as to avoid vaccine virus detection. After this grace period, if any result is positive, other molecular diagnostic methods such as ORF-5 or whole genome sequencing should be implemented to distinguish wild-type or vaccine PRRSV.

In parallel, in Europe, many questions have been raised during the last years regarding the safety and particularly the shedding and spreading of vaccinal strains, and consequently the risk of recombination between PRRS viruses (vaccinal or not) [7,8]. For this reason, the Committee for Medicinal Products for Veterinary Use (CVMP) of the European Medicines Agency (EMA) has included several warnings in the product information of all PRRS Modified Live Virus (MLV) vaccines used in Europe (EMA/324090/A/142).

Currently, few data are available regarding the capacity of different vaccinal virus strains to spread after vaccination. Recently, one case report failed to detect PRRSV-1 by qPCR on blood samples at weaning after a booster sow vaccination with ReproCyc^®^ PRRS EU [9]. In this cited study, in one herd, no vaccine virus was found in piglets in four successive batches weaned just after a booster sow vaccination.

In the present work, in five positive stable herds of different management practices and characteristics, we aimed to determine the frequency of detection of the vaccinal virus strain using qPCR on blood samples collected from piglets at weaning in batches weaned following a mass vaccination of their dams with ReproCyc^®^ PRRS EU (Boehringer Ingelheim, Ingelheim, Germany).

## 2. Materials and Methods

### 2.1. Study Design

This descriptive study was conducted in five commercial French pig herds located in Brittany. The characteristics of the herds included for the study are presented in Table 1.

The farms were enrolled between September 2021 to May 2022. All were presumed to be PRRSV positive stable and used ReproCyc^®^ PRRS EU in the sow herd.

According to the AASV guidelines [5], all of the farms were controlled positive stable before the study: blood samples were collected from one due-to-wean piglet per litter with a maximum of 30 piglets per batch over a 90-day period.

Then, a sow mass vaccination (SMV) was implemented the week just before a second monitoring phase, as previously described, started, with ReproCyc^®^ PRRS EU (2 mL, intra-muscular route—Boehringer Ingelheim, Ingelheim, Germany), using one sterile needle per sow.

### 2.2. Sample Collection

In each tested batch, the day before the weaning day, blood samples were collected from a convenient sample of one piglet per litter (in a maximum of 30 piglets per batch). The samples were collected in plain test tubes using one sterile needle per piglet from the cranial vena cava and were kept in cool storage (4 °C to 8 °C) until submission to the lab within 2 h after collection.

### 2.3. Diagnostic Testing

Diagnostic testing was performed at Labofarm (Finalab Veterinary Laboratories Group, Loudéac, France). The blood samples were centrifuged (4500× *g* for 5 min) and then the sera were tested for PRRSV RNA using Adiavet PRRSV real time 100R kit (BioX Diagnostics, Rochefort, Belgium). The serum samples were pooled by five as a maximum (or less if there were less than 30 litters in a batch). Positive and negative controls (RNA extracts from positive and negative samples, respectively) were included in each run as the quality control check. A sample was considered positive if the cycle threshold (Ct) value was ≤40 and the curve showed a specific exponential look.

### 2.4. Epidemiological Data Collection

A questionnaire with mainly semi-closed questions was developed to assess potential risks factors previously mentioned in the literature for the shedding of PRRSV strains on a production site. The questionnaire was filled out by the first author during a 1 h in-person interview with the farmer. Data on farm characteristics were recorded (pig inventory, batch management practice, weaning age, etc.). Data on specific external biosecurity measures implemented on the site were obtained (distance from pig sites in the neighborhood, deliveries of semen and gilts, and quarantine management). Finally, internal biosecurity measures that have an impact on PRRSV diffusion in swine herds were evaluated (pig and human flow, vaccination practices, hygiene and all-in all-out by room practices, gilts feedbacks in quarantine, cross fostering management, nursing sows, and batch mixing in gestation). When a biosecurity practice was reported as only partially applied, it was considered that the measure was not applied. For the five farms enrolled, the geographical distribution and PRRSV status of the pig sites within 5 km were obtained from the regional association dedicated to PRRSV surveillance (Organisation Sanitaire Porc Bretagne, Rennes, France). The questionnaire is available upon request from the first author.

## 3. Results

In all of the farms except Farm 3, 30 piglets were sampled at each sampling time. In Farm 3, the smallest farm in the study, between 17 to 28 litters were sampled each time, depending on the number of litters in the batch. In total, 886 piglets were sampled, which represented 229 pools tested by PCR in total.

All of the samples collected before the sow mass vaccination implemented during the study period were negative, confirming that all farms were considered as PRRS stable. In addition, no clinical sign suggesting PRRS infection was observed.

The timeline of the blood collections after SMV differed from farm to farm. It depended on the batch management system, the week of vaccination, and the first weaning time after SMV. All of the results by sampling time are summarized in Table 2. The time interval between the SMV and the first sampling and between SMV and the fourth sampling differed from 2 to 14 days and from 25 to 91 days, respectively. All of the results were negative in all batches in all farms. The two testing points listed the same week of SMV were implemented 2 days after vaccination.

The details of the farm audit are presented in Table 3. Some important characteristics regarding gilts management (gilts purchased in a multiplier, all-in all-out and duration of quarantine, age at delivery, and vaccination with two doses of a MLV during acclimatation), cleaning and disinfection protocols in quarantine and farrowing rooms, SMV scheme, and change of needle between sows were common between all farms. On the contrary, some practices differed from farm to farm: batch management system and age at weaning as previously noticed, mix of gestating sows and gilts in different pens or rooms, hygiene procedures at processing, gestion of human flow between physiological sectors, age maximum at adoption, and percentage of piglets adopted.

## 4. Discussion

This descriptive study was conducted in five herds with different production characteristics and management practices. No PRRSV-1 was detected by sampling 30 due-to-wean piglets born from sows vaccinated with ReproCYC^®^ PRRS EU from 2 to 91 days before piglet sampling, which confirm previous results [9]. These investigations would have allowed us to detect horizontal virus transmission (piglets born before SMV and in lactation phase at the moment their dams were vaccinated in farrowing rooms) as vertical (transplacental dissemination) when sampling piglets born after the vaccination of their mothers in gestation.

This absence of PRRSV detection in piglets even a few days after the vaccination of their dams could be unexpected. Cano et al. previously demonstrated that naïve sows’ PRRSV-2 infections in late gestation led to the birth of viraemic piglets [10]. Moreover, the summaries of the product characteristics of PRRS MLV vaccines available in Europe give the information that PRRSV-1 vaccinal strain could be detected in newborn piglets born to sows vaccinated at the end of gestation. This is also the reason AASV have introduced a minimum grace period before testing population after a SMV in the revised classification [6].

This absence of detection could be due to the sampling procedure we adopted. In this study, we focused only on piglets’ blood samples, which is representative of field conditions, and we did not combine it with other sample types. A larger sample size or a combination with other sample types such as processing fluids or family oral fluids would have allowed for detecting prevalence lower than 10% with a 95% confidence interval [6]. However, it has to be noticed that, in a previous study, our team already showed the absence of detection of the same vaccinal virus strain after sow vaccination, even when combining the blood samples of weaning piglets and processing fluids [9]. Parallelly, we pooled the samples by five as it is usually done this way in the field all around the world when monitoring PRRSV circulation in farrowing rooms. It is well know that this method could lower the detection rate of the virus, but to a lower extent, as it has been previously described [11].

Looking specifically at this isolate, 94,881 from ReproCyc^®^ PRRS EU, a previous report mentioned the lack of detection of this vaccine strain in the blood samples of sows after their vaccination [12]. More recently, Bourry et al. evaluated the viraemia and transmission capacities of different vaccine strains in SPF pigs after vaccination [13]. The results of both studies are consistent with ours, demonstrating a very low level and duration of viraemia of isolate 94,881 in vaccinated piglets and particularly no transmission to naïve piglets in direct contact with vaccinated ones.

In order to investigate if this observation should be done with other PRRSV-1 vaccinal strains, other studies should be conducted with the same protocol, but using different PRRSV-1 vaccines in different herds of various production characteristics and farm management practices. Indeed, in a Spanish study in more than 30 breeding herds, the authors succeeded in detecting PRRSV-1 through qPCR in blood samples of due-to-wean piglets following a SMV, with a different vaccinal virus strain than the one we used, in 13.8% of the cases [14]. Even if the authors did not sequence the virus, they suggested that it was the vaccine strain that infected the piglets.

Finally, it is noticeable that recombination events between PRRSV-1 vaccine strains are widely investigated and described, these events can sometimes lead to dramatic consequences [7,8]. To limit this risk of recombination, PRRS MLV vaccine transmission should be kept at a low level. In addition, the transmission capacity of each vaccine should be investigated in different conditions, as we did in this study.

In this study, we investigated the transmission capacity of only one MLV PRRSV isolate in five different herds. We selected farms in order to be representative of French production, with different sizes, different batch management systems, and age at weaning for example. In addition, the internal biosecurity measures applied in the gestating and farrowing units were quite variable and seemed to not influence PRRSV detection in the conditions of our study.

## 5. Conclusions

In this study, in five PRRSV-1 stable herds of different characteristics and management practices, we failed to detect the PRRSV-1 vaccine virus strain in piglets weaned after a booster vaccination in sows with ReproCyc^®^ PRRS EU. It confirms a previous report in only one herd and should mean that, with this vaccine strain, dissemination of the vaccine virus is a rare event. Nevertheless, the prudent use of MLV1s remains appropriate. The results we have generated should be confirmed and completed by further studies including other vaccines available on the European market.

## Figures and Tables

**Table 1 vetsci-10-00294-t001:** Studied farm’s characteristics.

No. Farms	1	2	3	4	5
Production system ^1^	FTW	FTF	FTF	FTF	FTW
Number of sows	1000	550	210	330	1000
Batch interval (weeks)	1	3	3	4	2
Batch management system (Number of batches)	20	7	7	5	10
Mean number of sows at farrowing per batch	48	70	26	62	90
Weaning age (days)	21	28	28	21	21

^1^ FTW: farrow-to-wean farm; FTF: farrow-to-finish farm.

**Table 2 vetsci-10-00294-t002:** Summary of PCR results in all of the farms during the study.

Week	−15	−14	−13	−12	−11	−10	−9	−8	−7	−6	−5	−4	−3	−2	−1	SMV	+1	+2	+3	+4	+5	+6	+7	+8	+9	+10	+11	+12	+13
Farm 1												6	6	6	6		6	6	6	6									
Farm 2							6			6			6			6		6		6			6			6			
Farm 3				5			6			6			4				4			5			4			5			
Farm 4	6				6				6				6				6				6				6				6
Farm 5										6		6		6		6			6		6		6		6				

Farms are displayed in rows and study weeks in columns. Cell codes: number of pools tested per sampling day; cell colors: green cells: negative PCR result; (SMV): sow mass vaccination applied (in week 0).

**Table 3 vetsci-10-00294-t003:** Farm information records based on audit questionnaire answers.

		Farm 1	Farm 2	Farm 3	Farm 4	Farm 5
Gilts origin and quarantine management	Rythm of deliveries (in weeks)	6	6	7	4	6
	Number of gilts per delivery	48	20	10	10	60
	Gilts origin? ^1^	C	C	C	C	C
	PRRSV status?	Naive	Naive	Naive	Naive	Naive
	N different ages from the nucleus herd	3	2	1	1	2
	Ages at delivery (in days)	140/180	160/180	180	168	155/175
	All in-all out (AIAO) management?	Yes	Yes	Yes	Yes	Yes
	If yes,					
	Duration (weeks)	6	11	6	10	6
	1-phase or 2-phases ^2^	1	2	1	1	2
	Gilts feed-back material in quarantine	feces	feces	feces	feces	feces
	N * Reprocyc^®^ PRRS EU vaccine injections in quarantine	2	2	2	1	2
	N * Reprocyc^®^ PRRS EU vaccine injections before AI *	2	2	2	1	2
Semen origin	Commercial Boar stud (PRRSV negative)?	Yes	Yes	Yes	Yes	Yes
Management of sows in gestation barn	Are the gilts in separated pens in gestation?	Yes	Yes	No	Yes	No
	Are groups fixed or dynamic?	Fixed	Fixed	Fixed	Fixed	Fixed
	Number of sows per pen in average	8	10	30	7	10
Management in farrowing rooms	AIAO per room?	Yes	Yes	Yes	Yes	Yes
	Use of nursing sows?	Yes	Yes	Yes	Yes	Yes
	% of gilts introduced during 3-months before SMV (mean per batch in %)	17	16	20	16	19
	% of gilts introduced during 3-months after SMV (mean per batch in %)	17	16	15	16	19
	Minimum age at fostering (in hours)	12	12	6	12	6
	Maximum age at fostering (in days)	5	8	10	8	8
	% of piglets fostered (mean per batch in %)	15	20	30	20	10
	Piglets stay in their pen when processing?	No	No	Yes	Yes	Yes
	Disinfection of material and change of needles between litter?	Yes	Yes	Yes	Yes	Yes
	PRRS vaccination of piglets	No	No	No	No	No
Internal Biosecurity/ Farmworkers	Dedicated farmworkers per physiological stage?	Yes	Yes	No	Yes	Yes
	Change of boots between sectors?	No	Yes	No	Yes	No
	Change of fomites between sectors?	No	Yes	No	Yes	No
Vaccination practices	Mass vaccination (all batches and animals the same day)?	No	Yes	Yes	No	Yes
	N batches postponed for vaccination	2	0	0	1	0
	Interval between each mass vaccination (in weeks)	16	15	16	16	16
	Heat detection boars’ vaccination?	Yes	Yes	Yes	Yes	Yes
	One needle per sow?	Yes	Yes	Yes	Yes	Yes

* N means “Number of.” and AI refers to artificial insemination; ^1^ Gilts origins can be commercial multiplier (C) or self-produced (S); ^2^ Observation phase and acclimatation phase in separate rooms with All-in All-out management.

## Data Availability

The study did not report any data.

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
