# Peer review of "PRRSV Detection by qPCR on Serum Samples Collected in Due-to-Wean Piglets in Five Positive Stable Breeding Herds Following a Sow Mass Vaccination with a Modified Live Vaccine: A Descriptive Study"

_vetsci, 2023, doi:10.3390/vetsci10040294_

Round 1

Reviewer 1 Report

The submitted manuscript by Lebret et al. describes a study on the detection of PRRSV in 5 herds undergoing a control program based on the use of MLV. The authors proved that in tested herds, despite mass vaccination of sows, no vaccine virus could be detected. The results are of interest due to an increasing interest in the control of PRRSV, which is one of the most economically important endemic viruses affecting pig production, but also in the light of recent reports on recombinations of vaccine strains with possible reversion to virulence.
The paper is well-written and properly organized. The detailed description of herds management and biosecurity practices is a strength of the study.

Minor comments:
- I miss some relevant background information about PRRSV transmission in the introduction, especially regarding factors influencing virus circulation in the herd and determining the control/diagnostic steps, like transplacental transmission or the significance of gilts acclimation.

- In Table 2, two batches were examined in the same week as sow mass vaccination was performed, but it is not specified if before or after SMV.

- Two explanatory notes under table 3 have the same footnote No. (1)

- Lines 156-157: The authors stated that they did not detect vaccine virus from piglets "even a few days after vaccination". This statement can be misleading, as the samples used in the study were collected from weaned piglets, so the results reported within 21-28 days after SMV originated from litters delivered before SMV. Based on these results the possibility of transmission of vaccine virus to the offspring can not be excluded, and the probability of detection would be higher based on sampling within a short time window after delivery e.g. testing of processing fluids from litters shortly after SMV.

- The ReproCyc PRRS writing should be unified throughout the text.

- The supplementary materials are listed, but there is no reference in the text.

Author Response

Dear reviewer, First of all, than you very much for your detailed reviewed and interesting comments.

Answers to minor comments :

Minor comments:
- I miss some relevant background information about PRRSV transmission in the introduction, especially regarding factors influencing virus circulation in the herd and determining the control/diagnostic steps, like transplacental transmission or the significance of gilts acclimation.

We added a sentence and changed another one to try to answer your comment. L36-38

- In Table 2, two batches were examined in the same week as sow mass vaccination was performed, but it is not specified if before or after SMV.

Thank you. You are right. We added a sentence L130-131.

- Two explanatory notes under table 3 have the same footnote No. (1)

Well noticed, corrected.

- Lines 156-157: The authors stated that they did not detect vaccine virus from piglets "even a few days after vaccination". This statement can be misleading, as the samples used in the study were collected from weaned piglets, so the results reported within 21-28 days after SMV originated from litters delivered before SMV. Based on these results the possibility of transmission of vaccine virus to the offspring can not be excluded, and the probability of detection would be higher based on sampling within a short time window after delivery e.g. testing of processing fluids from litters shortly after SMV.

Thank you, you are totally right. We added a comment on this point Line 155-159. Regarding processing fluids, there is already a comment in the discussion part.

- The ReproCyc PRRS writing should be unified throughout the text.

Corrected

- The supplementary materials are listed, but there is no reference in the text.

No supplementary material, corrected

Author Response

Dear reviewer, Thank you for your valuable comment.

Please add additional information regarding the qPCR assay in the methods section such as confirming that appropriate positive control samples were run with each PCR batch to ensure the assay is working appropriately given there were no positive results detected in the herd, and some information regarding the sensitivity of the assay.

We added a sentence L99-100 regarding the internal control check.

Other considerations: Line 83: unclear what is meant by max of 30 piglets per litter (do you mean per batch?)

Well noticed, corrected

Reviewer 3 Report

This descriptive study in five commercial pig farms with different management practices showed safety of using modified live vaccine. Brief report is clearly processed, the results in two clear tables.

But testing is questionable. Serum samples were pooled by five as maximum. It is questionable whether pooling five samples is appropriate. What would be the result of pooling the 2 samples? The same?  I am not sure.

Author Response

Thank you for your valuable comments.

But testing is questionable. Serum samples were pooled by five as maximum. It is questionable whether pooling five samples is appropriate. What would be the result of pooling the 2 samples? The same?  I am not sure

We added a sentence in the discussion part to “explain” our choice. L180-183